# Heuristic Domain Adaptation

**Shuhao Cui**[1,2*] **Xuan Jin**[3] **Shuhui Wang**[1†] **Yuan He**[3] **Qingming Huang**[1,2,4]

[1]Key Lab of Intell. Info. Process., Inst. of Comput. Tech., CAS
[2]University of Chinese Academy of Sciences  [3]Alibaba Group  [4]Peng Cheng Laboratory
{cuishuhao18s, wangshuhui}@ict.ac.cn, {jinxuan.jx, heyuan.hy}@alibaba-inc.com
qmhuang@ucas.ac.cn

## Abstract

In visual domain adaptation (DA), separating the domain-specific characteristics
from the domain-invariant representations is an ill-posed problem. Existing meth-
ods apply different kinds of priors or directly minimize the domain discrepancy
to address this problem, which lack flexibility in handling real-world situations.
Another research pipeline expresses the domain-specific information as a gradual
transferring process, which tends to be suboptimal in accurately removing the
domain-specific properties. In this paper, we address the modeling of domain-
invariant and domain-specific information from the heuristic search perspective.
We identify the characteristics in the existing representations that lead to larger
domain discrepancy as the heuristic representations. With the guidance of heuris-
tic representations, we formulate a principled framework of Heuristic Domain
Adaptation (HDA) with well-founded theoretical guarantees. To perform HDA,
the cosine similarity scores and independence measurements between domain-
invariant and domain-specific representations are cast into the constraints at the
initial and final states during the learning procedure. Similar to the final condi-
tion of heuristic search, we further derive a constraint enforcing the final range
of heuristic network output to be small. Accordingly, we propose Heuristic Do-
main Adaptation Network (HDAN), which explicitly learns the domain-invariant
and domain-specific representations with the above mentioned constraints. Ex-
tensive experiments show that HDAN has exceeded state-of-the-art on unsuper-
vised DA, multi-source DA and semi-supervised DA. The code is available at
https://github.com/cuishuhao/HDA.

## 1 Introduction

In visual domain adaptation scenarios, the domain-invariant (*a.k.a.* transferable) representations [36],
are considered to be the information bottleneck to enhance the generalizability of a model transferred
from labeled source domain to unlabeled or weakly-labeled target domain. Nevertheless, without
sufficient knowledge, separating domain-invariant information from the domain-specific information
that is harmful for domain adaptation, is substantially an ill-posed problem with infinite solutions.

To derive feasible solutions, a major way is to impose different prior knowledge on the structure of the
mapping functions. For example, the domain-invariant feature is regarded as the information shared
among domains, and obtained by averaging the domain-specific metrics [37] or by a joint neural
network [8] for multiple learning tasks. Regularization with well-established statistical explanation,
such as $F$-norm [15], structured $L_1$-norm [3, 4] or orthogonality [2] constraints on linear or kernel
mapping functions [14], is imposed on either the domain-specific or domain-invariant part to alleviate

the ill-posedness. However, the learning of domain-invariant representations may still suffer from inappropriate prior knowledge, which requires both a comprehensive understanding of domain data and the flexibility in handling real-world domain adaptation problems.

In deep neural networks, the learning of domain-invariant features are directly guided by knowledge of prior distribution using diverse metrics on multiple layers [17, 18, 32, 55, 59], or by confusing the domain classifier in an adversarial manner [18, 25, 28, 33]. Some other prior-based approaches are focused on additional properties along with the training procedure, such as the transferability [10, 54] and discriminability [10, 11, 60]. Nevertheless, by disregarding the explicit modeling of domain-specific characteristics, it can not be guaranteed that the domain-invariant representations contain as few domain-specific properties as possible. Other research pipeline expresses the domain-specific information from source to target as a gradual transferring process [12], undertaken on a geodesic domain flow along the data manifolds [19, 20, 23]. In these methods, the geodesic transfer properties are enforced by reconstructing input images [6, 20], which tends to be less accurate in removing the domain-specific properties from domain-invariant representations.

In this paper, we propose a principled framework of Heuristic Domain Adaptation (HDA), with a lower error bound derived by theoretical analysis. The information in the current domain-invariant representations that leads to larger domain discrepancy is identified as the domain-specific counterpart. We assume that domain-specific representations could be more easily obtained compared with domain-invariant parts. Similar to the estimated distance in heuristic search, the explicit construction of domain-specific representations could provide heuristics on learning the domain-invariant representations. Furthermore, the distance to the destination could be approximated by the range of domain-specific representations. Similar to the admissible constraint in heuristic search, the range is expected to be lower than the ideal range of domain-specific representations during the training. In what follows, we sometimes refer to the domain-specific representation as the heuristic representation.

To perform HDA, we derive three heuristic constraints, *i.e.*, limited similarity, independence and termination. The analysis on cosine similarity scores between domain-invariant and domain-specific representations results in the constraint of limited similarity between their initial representations. The independence between the domain-invariant and domain-specific representations could be enforced by lessening the non-gaussianity discrepancy between the two representations. We also analyze the termination condition of heuristic search, and derive the constraint of reducing the range of heuristic representations to near zero at the end. By distinguishing the two counterparts more comprehensively, a better construction of domain-invariant representation can be achieved.

Accordingly, we propose Heuristic Domain Adaptation Network (HDAN), which is composed of the fundament network and heuristic network. The heuristic network is constructed by a series of subnetworks to enhance the construction of domain-specific properties. We devise HDAN for three typical domain adaptation tasks, *i.e.*, unsupervised DA, semi-supervised DA and multi-source DA. Experiments show that HDAN achieves state-of-the-art results on the three tasks. Our contributions could be summarized as follows:

- We propose HDA, a principled framework leveraging domain-specific representations as heuristics to obtain domain-invariant representations.
- We develop three heuristic constraints to distinguishing the two counterparts more comprehensively, and a better construction of domain-invariant representation can be achieved.
- We design HDAN to perform HDA, and achieve state-of-the-art on three DA tasks.

## 2 Related Work

Visual domain adaptation [42] has achieved increasing research interests in the past few years. Recently, domain adaptation methods focus on different types of domain adaptation tasks, such as unsupervised DA [18, 54, 11], multi-source DA [53, 39], semi-supervised DA [43], partial DA [7] and unsupervised open domain recognition [58]. In this work, we aim to construct a general framework that is suitable for a wide range of domain adaptation tasks.

Deep domain adaptation methods incorporate various learning criteria on deep networks with two main technologies for distribution matching, *i.e.*, moment alignment and adversarial training. Moment alignment methods [23, 32, 55] are designed with classical metrics, such as maximum mean discrepancy [32, 35], second-order statistics [48, 59] or other distance metrics calculated on task-specific

representations [26]. Inspired by the Generative adversarial networks (GAN) [21], adversarial learning has been successfully explored on various tasks including domain adaptation. Domain adversarial neural network (DANN) [18] confuses the domain classifier by the gradient reversal layer to reduce the domain shift. Other methods, such as ADDA [50], CyCADA [25], MCD [45], CDAN [33] and GVB [12], further put forward more advanced adversarial domain adaptation frameworks.

Beyond distribution alignment, other types of prior knowledge [8, 4] can also be applied. The prior knowledge is regarded as heuristic information to enhance the reinforcement learning that is used to address the transfer learning problem [9]. In domain adaptation, the transferability is enhanced by minimizing the $L_1$-norm of domain-specific representation in [54, 10]. Meanwhile, the discriminability [10, 60, 11] is modeled to avoid spurious predictions. In our study, the prior knowledge is the assumption that domain-specific properties could be more easily obtained compared to domain-invariant properties. Thus similar to the estimated distance in heuristic search, we explicitly construct domain-specific representations towards more accurate domain adaptation.

In existing methods, DSN [6] and DLOW [20] reconstruct the input images to enhance the domain-specific counterpart. However, the image reconstruction demands large domain properties, resulting in great difficulties in removing the domain characteristics. GVB [12] explicitly reduces the domain discrepancy, but it lacks accurate construction of domain-specific parts. In comparison, we model the domain-specific parts from the overall representations by checking the conditions of the similarity, independence and termination in the learning process, similar to the procedure of heuristic search.

## 3 Heuristic Domain Adaptation

In classic path search problems, the goal is to obtain the minimum-cost path. As a typical solution of path search, heuristic search estimates the distance to destination with lower time consumption. In a standard heuristic search process, $e.g.$, $A^*$ search, the $n$th node is given a cost $f(n)$. $f(n)$ is composed of two elements, $i.e.$, $g(n)$ as the cost from the start to the $n$th node, and $h(n)$ as the estimated cost from the $n$th node to the destination, as follows:

$$f(n) = g(n) + h(n). \tag{1}$$

In search problems, $A^*$ search is proved to be optimal [40, 38], if $h(n)$ is admissible. The admissible constraint means the value of $h(n)$ is smaller than the ideal estimation $h^*(n)$, $i.e.$, the real distance from the $n$th node to the destination should be restricted as follows:

$$h(n) \leq h^*(n). \tag{2}$$

In domain adaptation, it is difficult to directly obtain domain-invariant representations due to the large discrepancy between the training and testing data. Thus towards more comprehensive reduction of domain discrepancy, we resort to more flexible prior knowledge to mitigate the transfer difficulties.

**Assumption 3.1** *In a certain domain, the difficulty of constructing domain-invariant representations is more than that of constructing domain-specific representations.*

As widely accepted, images from the same domain demonstrate similar domain-specific properties. For example, images taken from foggy situation constitute the domain with vast expanse of whiteness [46]. Therefore, the domain properties could be easily captured, as validated in Figure 2.

The assumption is regarded as the general prior knowledge in domain adaptation. Accordingly, we construct a framework which is composed of a fundament network $F$, and a heuristic network $H$, as shown in Figure 1. The heuristic network aims to explicitly model the discrepancy from the current representations to the ideal domain-invariant representations, denoted as heuristic representations. Similar to Eqn. 1, with the input $x$, subtracting heuristic representations $H(x)$ from fundament representations $F(x)$ results in the transferable representations $G(x)$, as follows:

$$G(x) = F(x) - H(x). \tag{3}$$

According to Assumption 3.1, $H(x)$ could be more easily modeled than $G(x)$. Thus $H(x)$ could provide guidance to the construction of $G(x)$. Meanwhile, considering the admissible constraint in heuristic search in Eqn. 2, the heuristic representations could be similarly restrained as follows:

$$|H(x)| \leq |H^*(x)| = |F(x) - F^*(x)|, \tag{4}$$

where $H^*(x)$ and $F^*(x)$ are the ideal heuristic and fundament representations. The range of $H(x)$ should not be larger than the ideal value $H^*(x)$. This indicates that $H(x)$ should not over-estimate the distance $F(x)$ to $F^*(x)$. Since large distance is always accompanied by redundant domain properties, leading to the confusion between $G(x)$ and $H(x)$. The construction of $H(x)$ and $F(x)$ formulates the principled framework of Heuristic Domain Adaptation. More discussions between heuristic search and HDA could be found in Section 1 in Supplementary.

To show the intrinsic mechanism of HDA, we follow the theory of [5]. Consider the source and target distribution $S$ and $T$ over a family of classifiers $F$ in hypothesis space $\mathcal{H_H}$. $\epsilon_S(F) = \mathbb{E}_{(x,y)\sim S}|F(x) - y|$ is the risk of a hypothesis $F \in \mathcal{H_H}$ in distribution $S$. $\epsilon_S(F_1, F_2) = \mathbb{E}_{(x,y)\sim S}|F_1(x) - F_2(x)|$ represents the disagreement between hypotheses $F_1, F_2 \in \mathcal{H_H}$ in distribution $S$. The notations in distribution $T$ are similar. We denote the ideal hypothesis as $F^* = \arg\min_F \epsilon_S(F) + \epsilon_T(F)$. The probabilistic bound [5] of the target risk $\epsilon_T(F)$ of hypothesis $F$ is given by the source risk, ideal joint hypothesis risk and domain discrepancy risk, as follows:

$$\epsilon_T(F) \leqslant \epsilon_S(F) + [\epsilon_S(F^*) + \epsilon_T(F^*)] + |\epsilon_S(F, F^*) - \epsilon_T(F, F^*)|. \tag{5}$$

For heuristic search, we have $F = G + H$, where $F$, $G$ and $H$ belong to the same hypothesis space $\mathcal{H_H}$. Also the goal of $F$ and $G$ is the same, thus the ideal hypothesis $F^* = G^*$. On the functional path towards the ideal hypothesis, the heuristic function $H$ models domain-specific parts on both $S$ and $T$. The domain-specific parts could also be approximately calculated by the parts in $F$ but not in $F^*$. The lower error bound in normal situation is proved in Section 2 in Supplementary. Typically, the two functions modeling domain-specific properties could be approximately regarded as positively correlated, as follows:

$$H = k(F - F^*) \qquad k \in (0, 1], \tag{6}$$

where $k$ is the positive correlation coefficient between $H$ and $F - F^*$, lower than 1 constrained by Eqn 4. Substituting $H$ with $F - G$, we could obtain the relationship between $G$ and $F$ as follows:

$$(1 - k)(F - F^*) = (G - G^*), \tag{7}$$

where the equation could be achieved on distribution of both $S$ and $T$. Then the risk $\epsilon(G, G^*) = (1 - k)\epsilon(F^* - F)$ is occupied on distributions of $S$ and $T$. Meanwhile, since both $G$ and $F$ could classify source samples correctly, the differences on source domain are small, which means $\epsilon_S(G) = \epsilon_S(F)$. Then we could calculate the target risk $\epsilon_T(G)$ as follows:

$$\begin{aligned} \epsilon_T(G) &\leqslant \epsilon_S(G) + [\epsilon_S(G^*) + \epsilon_T(G^*)] + |\epsilon_S(G, G^*) - \epsilon_T(G, G^*)| \\ &\leqslant \epsilon_S(F) + [\epsilon_S(F^*) + \epsilon_T(F^*)] + (1 - k)|\epsilon_S(F, F^*) - \epsilon_T(F, F^*)| \end{aligned} \tag{8}$$

where $k \in (0, 1]$, thus $1 - k \in [0, 1)$. The utilization of $G$ could restrain the error upper-bound to a lower value, demonstrating the superiority of HDA.

## 4 Heuristic Domain Adaptation Network

### 4.1 Heuristic Constraint

In domain adaptation, the heuristic representations are supposed to estimate the divergence to the ideal domain-invariant representations by constructing domain-specific information. Methods in [6, 20] calculate domain-specific parts by reconstructing input images. The range of the domain-specific part has not been restrained, leading to confused domain-specific and domain-invariant properties. We construct the basic network performing HDA from the perspective of representation separation. Accordingly, we analyze the similarity, independence and termination conditions of the two part of representations.

First, to construct the heuristic function, we analyze the similarity between domain-invariant representations $G(x)$ and domain-specific representations $H(x)$. In high dimensional space, cosine similarity [47, 31] is a common method to reflect the relationship between the representations in deep networks. The cosine similarity between $G(x)$ and $H(x)$ could be calculated as:

$$cos(\theta) = \frac{G(x) \cdot H(x)}{|G(x)|\,|H(x)|}, \tag{9}$$

where $cos(\theta) \in [-1, 1]$. We aim to eliminate the domain-specific parts in $G(x)$ with assistance of $H(x)$. $H(x)$ should not model any domain-specific representations, but $H(x)$ should construct the characteristics confused into $G(x)$ that are actually domain-specific parts. Thus $G(x)$ and $H(x)$ should contain similar domain-specific representations that could be achieved by making the initial

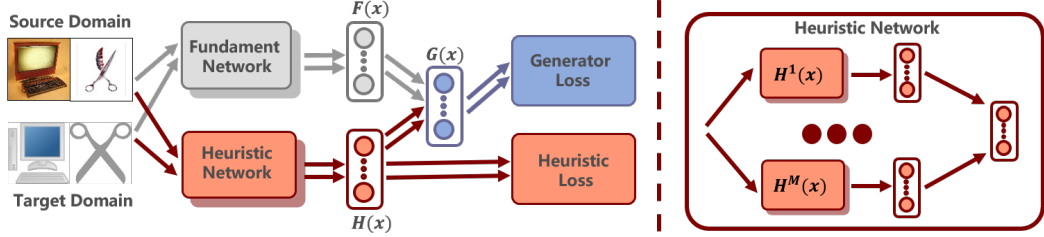

Figure 1: The proposed framework of Heuristic Domain Adaptation Network. The framework consists of a fundament network and a heuristic network. The heuristic network is further enhanced by multiple subnetworks modeling domain-specific representations. The overall loss consists of constraints on both the generator and heuristic loss functions.

cosine similarity to $1$ or $-1$. For better reducing the domain-specific parts, the initial similarity should be the minimum value $-1$. As the simplest form of minimum, we use the opposite initialization between $H(x)$ and $G(x)$. Denoting the initial value of $H(x)$ and $G(x)$ as $H_{init}(x)$ and $G_{init}(x)$, the heuristic similarity could be calculated as follows:

$$H_{init}(x) = -G_{init}(x). \tag{10}$$

Next, we analyze the independence relationship between $G(x)$ and $H(x)$. $G(x)$ should maintain the same across different domains, while $H(x)$ is different across domains. Thus $G(x)$ and $H(x)$ should be independent. Traditionally, a typical way to separate the statistically independent components is Independent Component Analysis (ICA) [1]. ICA is based on the Central Limit Theorem, *i.e.*, the distribution of a sum of independent random variables tends to converge to a Gaussian distribution.

Different from the ICA which learns the weight of each independent component, we aim to achieve a better separation from the existing representations $F(x)$. Specifically, as the sum of two independent variables produced by $G(x)$ and $H(x)$, $F(x)$ could obtain higher nongaussianity compared with $G(x)$ or $H(x)$. We choose a classical measurement of the nongaussianity, *i.e.*, Kurtosis as follows:

$$\text{kurt}(y) = \mathbb{E}_{y \in \mathcal{D}}\left[N(y)^4\right] - 3\left\{\mathbb{E}_{y \in \mathcal{D}}\left[N(y)^2\right]\right\}^2 \tag{11}$$

where $y$ is the variable sampled from $\mathcal{D}$ distribution, $\mathbb{E}$ is the expectation. $N(\cdot)$ is the process of normalization, resulting in distribution with zero mean and unit variance.

ICA maximizes the nongaussianity of the weighted sum, to obtain one of the independent components. Similarly, we could maximize the nongaussianity of $F(x)$, pushing $F(x)$ to be independent variable to obtain more accurate $G(x)$ and $H(x)$. Besides, we construct more accurate $G(x)$ by reducing the discrepancy between $G(x)$ and the ideal separation from $F(x)$. To approximate the ideal separation, we propose heuristic independence, a constraint reducing the nongaussianity discrepancy between $F(x)$ or $G(x)$. The heuristic independence could be calculated by the difference reduction between $\text{kurt}(F(x))$ and $\text{kurt}(G(x))$, as follows:

$$\text{kurt}(F(x)) - \text{kurt}(G(x)) \approx 0. \tag{12}$$

Finally, we re-investigate the termination condition of heuristic search. The path search terminates when the path reaches the destination with zero heuristic cost. On the path to the destination, the heuristic loss gradually reduces. Similarly, in domain adaptation, the range of heuristic representations $H(x)$ converges to near zero, if the constructed domain-invariant representations are converged to the ideal. In the training process, the range of heuristic representations should be progressively reduced to near zero towards more domain-invariant representations. Meanwhile, the heuristic termination reduces the range of $H(x)$ to be lower than the ideal, thus the termination constraint could also be regarded as the approximation of the constraint in Eqn. 4.

In some cases, the range of heuristic representations might be large for hard samples with rich domain-specific properties during the training process and also in the end. Considering that difficult samples hardly exist, the heuristic representations with large ranges in the dataset should be sparse. A common solution to ensure the sparsity is to minimize the $L_1$-norm [49]. The domain characteristic should be gradually reduced by penalizing large $L_1$-norm:

$$|H(x)|_1 \approx 0. \tag{13}$$

## 4.2 Heuristic Loss

For better explanation of the framework, we take the unsupervised domain adaptation for example. We are given source domain $\mathcal{D}_S = \{(x_i^s, y_i^s)\}_{i=1}^{N_s}$ with $N_s$ labeled examples covering $C$ classes where $y_i^s \in \{1, ..., C\}$ and target domain $\mathcal{D}_T = \{x_i^t\}_{i=1}^{N_T}$ with $N_t$ unlabeled examples that belong to the same $C$ classes. Example from either domain is represented as $x_i$.

For domain adaptation, the total loss $\mathcal{L}_F$ is composed of two parts, *i.e.*, the generator loss $\mathcal{L}_G$ and the heuristic loss $\mathcal{L}_H$. The generator loss could be calculated by the sum of classification loss $\mathcal{L}_{Cls}$ and transfer loss $\mathcal{L}_{Trans}$, as follows:

$$\mathcal{L}_F = \mathcal{L}_G + \mathcal{L}_H = \mathcal{L}_{Cls} + \mathcal{L}_{Trans} + \mathcal{L}_H. \tag{14}$$

On the generator, we directly calculate the adversarial discrepancy on classification responses, the same as [45, 12]. Despite of using the adversarial discrepancy, we should emphasize that HDA can also collaborate well with other discrepancy measurements. We denote $L_{ce}$ as the cross-entropy loss function and $D$ as the constructed discriminator. Then the classification loss and the transfer loss could be calculated as:

$$\mathcal{L}_{Cls} = \mathbb{E}_{(x_i^s, y_i^s) \sim \mathcal{D}_S} L_{ce}(G(x_i^s)), y_i^s)$$
$$\mathcal{L}_{Trans} = -\mathbb{E}_{x_i^s \sim \mathcal{D}_S} \log \left[ D(G(x_i^s)) \right] - \mathbb{E}_{x_j^t \sim \mathcal{D}_T} \log \left[ 1 - D(G(x_j^t)) \right]. \tag{15}$$

For the heuristic loss $\mathcal{L}_H$, we should not only restrain $H(x_i)$ to be domain-specific, but also satisfy the constraints discussed in Sec. 4.1. It is difficult to directly restrain $H(x)$ to model the domain-specific properties, while alignment between $G(x_i^s)$ and $G(x_i^t)$ could contribute to more domain properties in $H(x_i)$, as discussed in [6, 12]. To enhance the ability in capturing domain-specific properties, we design $H(x_i)$ to be an additive ensemble of multiple subnetworks:

$$H(x_i) = \Sigma_{k=1}^M H^k(x_i), \tag{16}$$

where $M$ is the number of heuristic subnetworks. Each heuristic subnetwork $H^k(x_i)$ is differently initialized. Thus each $H^k(x_i)$ models the local domain-specific property, providing more subtle fitting ability to the domain-specific properties for the whole heuristic network $H(x_i)$, which is validated by experimental results in Figure 3. With the combination of multiple subnetworks, the domain-specific properties could be captured more accurately by $H(x_i)$ with larger $M$. A larger number of subnetworks $M$ may lead to more accurate construction of heuristic representations, and also higher risk of over-fitting. The proper number of heuristic subnetworks is shown in Table 2.

For heuristic similarity, considering that the summation of $H(x_i)$ and $G(x_i)$ is $F(x_i)$, the constraint in Eqn. 10 becomes that $F(x_i)$ should be near zero for initialization. Thus the heuristic similarity is achieved by setting the initial fundament network parameters to near zero. To satisfy constraints of heuristic independence and termination, the range of heuristic representations should be reduced, similar to the constraint in GVB [12]. Since the constructed heuristic function contains multiple heuristic subnetworks, the reduction of range is achieved by minimizing the following objective function:

$$\mathcal{L}_H = \mathbb{E}_{x_i \sim (\mathcal{D}_S, \mathcal{D}_T)} |H(x_i)|_1 = \mathbb{E}_{x_i \sim (\mathcal{D}_S, \mathcal{D}_T)} \left| \Sigma_{k=1}^M H^k(x_i) \right|_1. \tag{17}$$

In Figure 4, the termination constraint is shown to have similar evolving pattern to the independence constraint of Eqn. 12 along the iteration number. The phenomenon is normal since a smaller range of $H(x_i)$ could lead to smaller difference between $G(x_i)$ and $F(x_i)$, which ensures similar nongaussianity on the other hand.

## 5 Experiments

We evaluate our method on three challenging domain adaptation tasks, *i.e.*, unsupervised domain adaptation, multi-source domain adaptation and semi-supervised domain adaptation. For the three tasks, we construct similar frameworks pre-trained on ImageNet [13], implemented with PyTorch. The heuristic network and the fundament network share the same feature extractor to reduce the number of parameters. To perform adversarial learning, we apply gradient reversal layer (GRL) [17] to the network. At the early stages of the training procedure, the progressive training strategy in GRL could avoid noisy signal from the domain classifier. Besides, we utilize entropy conditioning [33] for alignment between easy-to-classify samples. We employ Stochastic Gradient Descent (SGD) with a momentum of 0.9 and a weight decay of 0.0005 to train our model. We repeat the experiments of our proposed method over 3 times to report the average of the top-1 classification accuracy. For fair comparisons, the results are directly reported from their original papers if available.

Table 1: Accuracies (%) on Office-Home for ResNet50-based UDA.

| Method | A→C | A→P | A→R | C→A | C→P | C→R | P→A | P→C | P→R | R→A | R→C | R→P | Avg |
|---|---|---|---|---|---|---|---|---|---|---|---|---|---|
| ResNet50 [24] | 34.9 | 50.0 | 58.0 | 37.4 | 41.9 | 46.2 | 38.5 | 31.2 | 60.4 | 53.9 | 41.2 | 59.9 | 46.1 |
| DAN [32] | 43.6 | 57.0 | 67.9 | 45.8 | 56.5 | 60.4 | 44.0 | 43.6 | 67.7 | 63.1 | 51.5 | 74.3 | 56.3 |
| DANN [18] | 45.6 | 59.3 | 70.1 | 47.0 | 58.5 | 60.9 | 46.1 | 43.7 | 68.5 | 63.2 | 51.8 | 76.8 | 57.6 |
| MCD [45] | 48.9 | 68.3 | 74.6 | 61.3 | 67.6 | 68.8 | 57 | 47.1 | 75.1 | 69.1 | 52.2 | 79.6 | 64.1 |
| EntMin [52] | 43.2 | 68.4 | 78.4 | 61.4 | 69.9 | 71.4 | 58.5 | 44.2 | 78.2 | 71.1 | 47.6 | 81.8 | 64.5 |
| CDAN [33] | 50.7 | 70.6 | 76.0 | 57.6 | 70.0 | 70.0 | 57.4 | 50.9 | 77.3 | 70.9 | 56.7 | 81.6 | 65.8 |
| Symnets [56] | 47.7 | 72.9 | 78.5 | 64.2 | 71.3 | 74.2 | 63.6 | 47.6 | 79.4 | 73.8 | 50.8 | 82.6 | 67.2 |
| SAFN [54] | 52.0 | 71.7 | 76.3 | 64.2 | 69.9 | 71.9 | 63.7 | 51.4 | 77.1 | 70.9 | 57.1 | 81.5 | 67.3 |
| ATM [29] | 52.4 | 72.6 | 78.0 | 61.1 | 72.0 | 72.6 | 59.5 | 52.0 | 79.1 | 73.3 | 58.9 | 83.4 | 67.9 |
| BNM [11] | 52.3 | 73.9 | 80.0 | 63.3 | 72.9 | 74.9 | 61.7 | 49.5 | 79.7 | 70.5 | 53.6 | 82.2 | 67.9 |
| MDD [57] | 54.9 | 73.7 | 77.8 | 60.0 | 71.4 | 71.8 | 61.2 | 53.6 | 78.1 | 72.5 | 60.2 | 82.3 | 68.1 |
| GVBG [12] | 56.5 | 74.0 | 79.2 | 64.2 | 73.0 | 74.1 | 65.2 | 55.9 | 81.2 | 74.2 | 58.2 | 84.0 | 70.0 |
| CADA [27] | **56.9** | **76.4** | **80.7** | 61.3 | **75.2** | 75.2 | 63.2 | 54.5 | 80.7 | 73.9 | **61.5** | 84.1 | 70.2 |
| HDAN | 56.8 | 75.2 | 79.8 | **65.1** | 73.9 | 75.2 | **66.3** | **56.7** | **81.8** | 75.4 | 59.7 | **84.7** | **70.9** |

Table 2: Parameter $M$ and ablation study on Office-Home for HDAN.

| Method | A→C | A→P | A→R | C→A | C→P | C→R | P→A | P→C | P→R | R→A | R→C | R→P | Avg |
|---|---|---|---|---|---|---|---|---|---|---|---|---|---|
| HDAN (M=1) | 57.2 | 74.7 | 79.7 | **65.1** | 73.6 | 75.4 | 64.8 | 55.9 | 81.6 | 74.4 | 59.1 | 84.4 | 70.5 |
| HDAN (M=2) | **57.4** | 75.3 | 79.5 | **65.1** | 74.0 | 74.9 | 65.9 | 56.4 | 81.8 | 74.9 | **59.7** | 84.4 | 70.8 |
| HDAN (M=3) | 56.8 | 75.2 | **79.8** | **65.1** | 73.9 | **75.2** | 66.3 | 56.7 | 81.8 | **75.4** | **59.7** | **84.7** | **70.9** |
| HDAN (M=4) | 56.3 | **75.4** | 79.4 | 64.4 | **74.8** | 74.8 | **66.5** | 56.8 | **82.0** | 75.3 | **59.7** | 84.6 | 70.8 |
| HDAN (M=5) | 56.2 | 74.9 | 79.6 | 64.9 | 73.8 | 74.7 | 66.9 | 56.2 | 81.4 | 75.3 | 59.4 | 84.5 | 70.6 |
| HDAN $_{w.o.range}$ | 53.8 | 69.9 | 78.3 | 59.9 | 72.1 | 73.0 | 60.0 | 51.8 | 79.7 | 70.8 | 56.6 | 83.0 | 67.4 |
| HDAN $_{w.o.init}$ | 56.2 | 74.3 | 79.4 | 64.9 | 73.3 | 74.9 | 65.6 | 55.5 | 81.3 | 74.3 | 59.3 | 84.5 | 70.3 |
| HDAN $_{L_2-norm}$ | 56.6 | 74.9 | 79.9 | 64.6 | 73.7 | 74.3 | 65.8 | 56.1 | 81.4 | 74.9 | 59.4 | 84.4 | 70.5 |

## 5.1 Unsupervised Domain Adaptation

Unsupervised domain adaptation (UDA) is the classical domain adaptation task performed on labeled source domain and unlabeled target domain. In UDA, we use a standard dataset Office-Home [51] with 15,500 images in 65 categories. It has four significantly different domains: Artistic images (A), Clip art (C), Product images (P), and Real world (R). The four domains build up 12 challenging UDA scenarios in total.

The experiment results based on ResNet50 [24] are shown in Table 1. Experiments show that HDAN could achieve state-of-the-art results on Office-Home. On average, HDAN could achieve 0.9% improvements compared with the method of GVBG [12]. The improvement shows the importance of heuristic similarity and multiple heuristic subnetworks on Office-Home. Besides, HDAN outperforms competitors by a large margin, which shows HDAN is a more powerful framework in UDA.

We also show the results under different settings of $M$ and ablation study in Table 2. When $M = 1$, there is only one heuristic sub-network. HDAN achieves the best performance when $M = 3$ on Office-Home. This validates that more heuristic sub-networks could enhance the ability of modeling heuristics. If $M$ becomes too large, the heuristic function is difficult to optimize and the whole model suffers from over-fitting. Thus we fix $M = 3$ for consequent experiments. In ablation study, results of HDAN become worse without heuristic termination or heuristic similarity, which validates the positive effect of the constraints. HDAN with $L_2$-norm in Eqn. 17 performs worse than $L_1$-norm.

To show more domain-specific properties in $H(x)$, we construct two extra domain classifier to calculate the domain belongings in Figure 2, and the details are discussed in Section A.3 in Supplementary. Results show that the domain classification accuracy tends to be higher in $H(x)$ compared to $G(x)$. This result shows that more domain properties exist in $H(x)$. Meanwhile, each $H^k$ is initialized with random weights generated from Gaussian with different variance, resulting in different output ranges in the training process, as shown in Figure 3(a). It could also ensure that distinguished local or cluster-sensitive domain-specific properties can be captured by different $H^k$, as shown in Figure 3(b) and 3(c). The ensemble of $H^k(x)$ can capture rich domain-specific properties more effectively, evidenced by the low similarity between $H^k(x)$ and $G(x)$, *i.e.*, $\cos(G(x), H^k(x)) \approx 0$, as shown in Figure 3(b). Moreover, different $H^k(x)$ contains distinguished local property, since $H^k(x)$ and

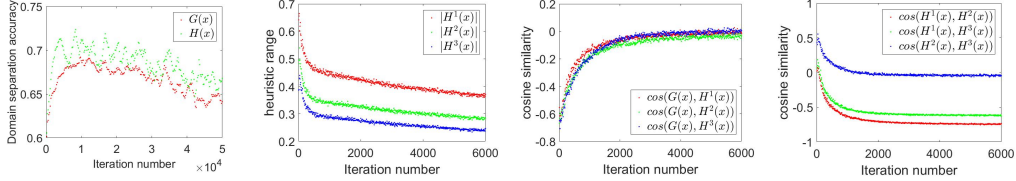

Figure 2: Domain accuracy (%) on C→R.

(a) $|H^k(x)|$       (b) $G(x)$ and $H^k(x)$       (c) $H^k(x)$ and $H^{k'}(x)$

Figure 3: Relationship among different domain parts on A→C.

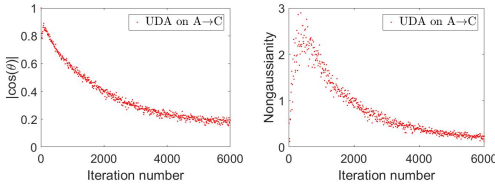

(a) Cosine similarity       (b) Nongaussianity

Figure 4: Losses of HDAN on A→C.

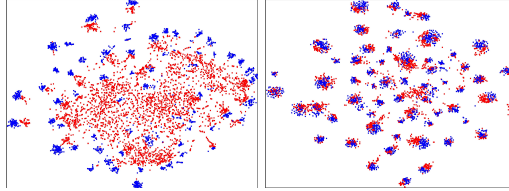

(a) ResNet50       (b) HDAN

Figure 5: T-SNE results of HDAN on A→C.

$H^{k'}(x)$ $(k \neq k')$ is orthogonal or negatively correlated, *i.e.*, $\cos(H^k(x), H^{k'}(x)) \leq 0$, as shown in Figure 3(c). Therefore, the aggregation of $H^k(x)$ could express more diversified domain-specific properties encoded by each sub-network.

We also visualize the loss functions in the training process of HDAN on A→C in Figure 4. As expected, the cosine similarity reduces rapidly. The nongaussianity gap between $G_n(x)$ and $H_n(x)$ is also progressively reduced. This validates that heuristic termination could ensure the independence between the domain-specific and domain-invariant representations. We also visualize the T-SNE results in Figure 5, with source domain in blue and target domain in red. The visualization shows that HDAN could achieve surprisingly good distribution alignment at the category level on Office-Home.

## 5.2 Multi-source Domain Adaptation

Multi-source domain adaptation (MSDA) is a challenging task, since the multiple source domains bring great difficulties to the domain adaptability. For evaluation on MSDA, we utilize a challenging dataset DomainNet [39], which contains about 600,000 images in 345 categories. DomainNet consists of six significantly different domains: Clipart, Infograph, Painting, Quickdraw, Real and Sketch. We follow the same data split scheme of training and validation images in the paper. In each domain adaptation setting, we choose only one domain as target domain, with others as the source domains. Thus the six domains formulate six challenging MSDA scenarios in total. To simplify, we only choose the target domain label as the label of different setting. For example, the label Clipart refers to the setting of transferring from Inforgraph, Painting, Quickdraw, Real and Sketch to Clipart.

Different from the setting of UDA, multiple source domains are labeled as different domains in adversarial training. Thus we make predictions on the domain labels at the discriminator side. On the generator, we keep fixing the number of heuristic network $M$ to 3, according to previous ablation study. We choose ResNet101 [24] as the basic backbone, and set the initial learning rate as 0.0003. Since methods trained under MSDA setting demonstrate large performance gap over methods trained under the single best setting as reported in [39], we only report results under the setting of combined source and multi-source in this paper. We reproduce GVBG [12] in the same settings with our method for fair comparison.

The experimental results of multi-source domain adaptation are shown in Table 3. Our method HDAN achieves state-of-the-art results on DomainNet. More specifically, HDAN outperforms other methods on all the settings except two challenging ones, *i.e.*, Quickdraw and Infograph. The results indicate that with the global guidance of heuristic information, HDAN is quite suitable for normal multi-source domain adaptation. HDAN also outperforms GVBG [12], which shows the effectiveness of our constraints. Besides, the standard deviations of our results are relatively small, which means the training process of HDAN performs smoothly.

## 5.3 Semi-supervised Domain Adaptation

Semi-supervised domain adaptation (SSDA) is a more realistic setting, where the domain adaptation is performed from labeled source domain to partially labeled target domain. In SSDA, we utilize the

Table 3: Accuracies (%) on DomainNet for ResNet101-based MSDA methods.

| Models | Clipart | Infograph | Painting | Quickdraw | Real | Sketch | Avg |
|---|---|---|---|---|---|---|---|
| ResNet101 [24] | 47.6±0.52 | 13.0±0.41 | 38.1±0.45 | 13.3±0.39 | 51.9±0.85 | 33.7±0.54 | 32.9±0.54 |
| DAN [32] | 45.4±0.49 | 12.8±0.86 | 36.2±0.58 | 15.3±0.37 | 48.6±0.72 | 34.0±0.54 | 32.1±0.59 |
| RTN [34] | 44.2±0.57 | 12.6±0.73 | 35.3±0.59 | 14.6±0.76 | 48.4±0.67 | 31.7±0.73 | 31.1±0.68 |
| JAN [35] | 40.9±0.43 | 11.1±0.61 | 35.4±0.50 | 12.1±0.67 | 45.8±0.59 | 32.3±0.63 | 29.6±0.57 |
| DANN [18] | 45.5±0.59 | 13.1±0.72 | 37.0±0.69 | 13.2±0.77 | 48.9±0.65 | 31.8±0.62 | 32.6±0.68 |
| ADDA [50] | 47.5±0.76 | 11.4±0.67 | 36.7±0.53 | 14.7±0.50 | 49.1±0.82 | 33.5±0.49 | 32.2±0.63 |
| SE [16] | 24.7±0.32 | 3.9±0.47 | 12.7±0.35 | 7.1±0.46 | 22.8±0.51 | 9.1±0.49 | 16.1±0.43 |
| MCD [45] | 54.3±0.64 | 22.1±0.70 | 45.7±0.63 | 7.6±0.49 | 58.4±0.65 | 43.5±0.57 | 38.5±0.61 |
| DCTN [53] | 48.6±0.73 | 23.5±0.59 | 48.8±0.63 | 7.2±0.46 | 53.5±0.56 | 47.3±0.47 | 38.2±0.57 |
| M$^3$SDA [39] | 57.2±0.98 | 24.2±1.21 | 51.6±0.44 | 5.2±0.45 | 61.6±0.89 | 49.6±0.56 | 41.5±0.74 |
| M$^3$SDA-$\beta$ [39] | 58.6±0.53 | 26.0±0.89 | 52.3±0.55 | 6.3±0.58 | 62.7±0.51 | 49.5±0.76 | 42.6±0.64 |
| ML-MSDA [30] | 61.4±0.79 | **26.2**±0.41 | 51.9±0.20 | **19.1**±0.31 | 57.0±1.04 | 50.3±0.67 | 44.3±0.57 |
| GVBG [12] | 61.5±0.44 | 23.9±0.71 | 54.2±0.46 | 16.4±0.57 | 67.8±0.98 | 52.5±0.62 | 46.0±0.63 |
| HDAN | **63.6**±0.35 | 25.9±0.16 | **56.1**±0.38 | 16.6±0.54 | **69.1**±0.42 | **54.3**±0.26 | **47.6**±0.40 |

Table 4: Accuracies (%) on DomainNet subset of ResNet34-based SSDA methods.

| Methods | R→C | | R→P | | P→C | | C→S | | S→P | | R→S | | P→R | | Avg | |
|---|---|---|---|---|---|---|---|---|---|---|---|---|---|---|---|---|
| | $1_{shot}$ | $3_{shot}$ | $1_{shot}$ | $3_{shot}$ | $1_{shot}$ | $3_{shot}$ | $1_{shot}$ | $3_{shot}$ | $1_{shot}$ | $3_{shot}$ | $1_{shot}$ | 3-shot | $1_{shot}$ | $3_{shot}$ | $1_{shot}$ | $3_{shot}$ |
| ResNet34 [24] | 55.6 | 60.0 | 60.6 | 62.2 | 56.8 | 59.4 | 50.8 | 55.0 | 56.0 | 59.5 | 46.3 | 50.1 | 71.8 | 73.9 | 56.9 | 60.0 |
| DANN [18] | 58.2 | 59.8 | 61.4 | 62.8 | 56.3 | 59.6 | 52.8 | 55.4 | 57.4 | 59.9 | 52.2 | 54.9 | 70.3 | 72.2 | 58.4 | 60.7 |
| ADR [44] | 57.1 | 60.7 | 61.3 | 61.9 | 57.0 | 60.7 | 51.0 | 54.4 | 56.0 | 59.9 | 49.0 | 51.1 | 72.0 | 74.2 | 57.6 | 60.4 |
| CDAN [33] | 65.0 | 69.0 | 64.9 | 67.3 | 63.7 | 68.4 | 53.1 | 57.8 | 63.4 | 65.3 | 54.5 | 59.0 | 73.2 | 78.5 | 62.5 | 66.5 |
| ENT [22] | 65.2 | 71.0 | 65.9 | 69.2 | 65.4 | 71.1 | 54.6 | 60.0 | 59.7 | 62.1 | 52.1 | 61.1 | 75.0 | 78.6 | 62.6 | 67.6 |
| MME [43] | 70.0 | 72.2 | 67.7 | 69.7 | 69.0 | 71.7 | 56.3 | 61.8 | 64.8 | 66.8 | 61.0 | 61.9 | 76.1 | 78.5 | 66.4 | 68.9 |
| GVBG [12] | 70.8 | 73.3 | 65.9 | 68.7 | 71.1 | 72.9 | 62.4 | 65.3 | 65.1 | 66.6 | 67.1 | 68.5 | 76.8 | 79.2 | 68.4 | 70.6 |
| CANN [41] | **72.7** | **75.4** | **70.3** | **71.5** | 69.8 | **73.2** | 60.5 | 64.1 | **66.4** | **69.4** | 62.7 | 64.2 | **77.3** | **80.8** | 68.5 | 71.2 |
| HDAN | 71.7 | 73.9 | 67.1 | 69.1 | **72.8** | 73.0 | **63.7** | **66.3** | 65.7 | 67.5 | **69.2** | **69.5** | 76.6 | 79.7 | **69.5** | **71.3** |

standard dataset proposed by MME [43], which is selected from DomainNet [39]. We take the same protocol as [43] where the four domains including Real (R), Clipart (C), Painting (P) and Sketch (S) in total 126 categories. In this paper, same as [43], seven domain adaptation settings are formulated on the four domains, showing different levels of domain gap.

We use the ResNet34 [24] as the backbones of the generator and fix the number of sub-networks to be 3, and with the initial learning rate as 0.001. On the discriminator, the labeled target domain is labeled as the third domain in the minmax game. All the methods are evaluated under the one-shot and three-shot settings following [43] where there are only one or three labeled samples per class in the target domain. We also reproduce the results of GVBG [12] in the same settings as our method, by fitting the adversarial process into semi-supervised domain adaptation.

The quantitative results on the selected DomainNet subset are summarized in Table 4. It is easily observed that HDAN outperforms all the competitors, and achieves state-of-the-art results on average. For example, compared with CANN, our method improves significantly on the difficult tasks, such as $R \rightarrow S$. HDAN also achieves significant improvement on one-shot SSDA, which shows the superiority of HDAN. Besides, the improvement compared with GVBG [12] shows the effectiveness of our heuristic constraints.

# 6 Conclusion

We consider to explicitly separate domain-specific characteristics from the domain-invariant representations for domain adaptation. Inspired by heuristic search, we propose a new framework HDA which achieves lower error bound by theoretical analysis. We design deep network HDAN to perform HDA, which contains multiple subnetworks in the heuristic network $H$ to ensure accurate modeling of domain-specific properties. Three constraints on initial parameter similarity, network output independence and the termination of heuristics are designed to train HDAN, which ensures more comprehensive reduction of domain properties. Experiments show state-of-the-art results achieved by HDAN on three domain adaptation tasks.

# 7    Broader Impact

Domain adaptation aims to obtain the domain-invariant representations across training and testing distributions, which is a general problem in machine learning. Our method stresses the importance of explicitly constructing domain-specific representations as the heuristics towards effective domain adaptation. The guidance of heuristics formulate the principled framework of HDA, similar to heuristic search. Despite of the similarity in working mechanism, remarkable difference still exist between heuristic search and our method from the task setting aspect.

In our method, heuristics are constructed in a way similar to blind signal separation. The separation process could also be applied to a broader range of tasks, such as signal reconstruction, representation learning or visual tracking. However, the separation could not achieve ideal separation without extra knowledge or priors. In common situations, there is no general method constructing the heuristic knowledge, which is another limitation of the method.

The framework of HDA could also be further improved from two perspectives. First, some simply constructed extra domain-specific knowledge could be introduced to enhance the separation. Moreover, the structure of our constructed heuristic networks is an additive ensemble of sub-networks, which could be replaced by Neural Architecture Search (NAS).

**Acknowledgement**. This work was supported in part by the National Key R&D Program of China under Grant 2018AAA0102003, in part by National Natural Science Foundation of China: 61672497, 61620106009, 61836002, 61931008 and U1636214, and in part by Key Research Program of Frontier Sciences, CAS: QYZDJ-SSW-SYS013.

## Footnotes

*Visiting student at Alibaba Group.

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
