[Supplementary Material]

# Heuristic Domain Adaptation

## A Appendix

### A.1 Main Differences

Table 1: Main differences between heuristic search and HDA.

| Property | Heuristic search | HDA |
|---|---|---|
| Goal | A least-cost path to destination | Domain-invariant representation |
| Way | Selected path | Network parameters |
| Number | The $n$th node | The $n$th iteration |
| Mechanism | Total cost $f(n)$ <br> Existing search cost $g(n)$ <br> Estimated heuristic cost $h(n)$ <br> $f(n) = g(n) + h(n)$ <br> $f(n+1) = min(g(n) + c(n,m) + h(m))$ | Fundament representation $F(x)$ <br> Generator representation $G(x)$ <br> Heuristic representation $H(x)$ <br> $G(x) = F(x) - H(x)$ <br> $H(x) = \Sigma_{k=1}^{M} H^k(x)$ |
| Termination | No path eligible to be extended | No more invariant representation |
| Admissible | $h(n) \leq h^*(n)$ | $\|H(x)\| \leq \|H^*(x)\|$ |
| Loss | $f(n) = g(n) + h(n)$ | $\mathcal{L}_F = \mathcal{L}_G + \mathcal{L}_H$ |

Heuristic search is devised for the task of path search problems, while heuristic domain adaptation (HDA) is implemented on the task of domain adaptation. The different tasks result in different properties between heuristic search and HDA. We summarize the main different properties as shown in Table 1.

Heuristic search aims to obtain a least-cost path to the destination. While the goal of HDA is to obtain the domain-invariant representations, in a goal-oriented manner. On the way to the destination, heuristic search is achieved by progressively selecting the extended path. Meanwhile, HDA gets closed to the destination by training the network parameters. During the training process, number $n$ denotes the $n$th node in path search problems, while number $n$ also means the iteration number in the training process.

The path cost in heuristic search is similar to the constructed representations in HDA. The cost of searching to the existing node $g(n)$ could be analogous to existing domain-invariant representations $G(x)$. As the other part, the cost of estimating the distance from node $n$ to the destination $h(n)$ could be similar to the domain-specific representations $H(x)$. Then the range of domain-specific representations can be regarded as the approximate distance from the current to the ideal solution. The fundament representations $F(x)$ could be calculated by the sum of $G(x)$ and $H(x)$. In path search problem, the total cost $f(n)$ is the sum of the given $g(n)$ and $h(n)$. While the calculation is slightly different from HDA, by the construction of $F(x)$ and $H(x)$ in domain adaptation. Besides, the node in path search is extended when the node extension could reach the optimal at the node of $m$, by calculating and selecting the least cost of the sum of $g(n)$, $c(n,m)$ as the cost from node $n$ to $m$, and $h(n)$ as the cost of estimated cost. While the costs of all the previous representations are difficult to calculate. We choose to enhance $H(x)$ by constructing multiple heuristic subnetworks.

In traditional path search problems, the path search could terminate when no path eligible to be extended or the destination is reached. In domain adaptation, the ideal domain-invariant representations are hardly to obtain, thus domain adaptation would terminate when no more invariant representations could be obtained. In path search, the ending constraints could be approximated by zero heuristic loss $h(n)$. Heuristic search could be optimal when admissible, which means $h(n) \leq h^*(n)$. Similarly, the range of heuristic representations $H(x)$ should be lower than the ideal value $H^*(x)$. In the final,

the range of $H(x)$ should also be near zero. To limit the representations, the loss functions in domain adaptation are similar to the path search costs of $f(n) = g(n) + h(n)$. Then the total loss functions consist of the generator loss and heuristic loss functions.

| (a) HDA | (b) HDA$_{w.o.init}$ |
|---------|----------------------|

Figure 1: Cosine Similarity on A→C.

| (a) HDA($M=3$) | (b) HDA($M=1$) |
|----------------|----------------|

Figure 2: Heuristic loss on A→C.

Table 2: Comparison with and without nongaussian on Office-Home for HDA.

| Method | A→C | A→P | A→R | C→A | C→P | C→R | P→A | P→C | P→R | R→A | R→C | R→P | Avg |
|--------|-----|-----|-----|-----|-----|-----|-----|-----|-----|-----|-----|-----|-----|
| HDA | **56.8** | 75.2 | 79.8 | **65.1** | 73.9 | **75.2** | 66.3 | 56.7 | **81.8** | 75.4 | **59.7** | **84.7** | **70.9** |
| HDA $_{nongaussian}$ | 56.5 | **75.3** | **79.9** | **65.2** | **74.0** | 75.0 | 66.0 | **56.8** | 81.3 | 74.5 | 59.6 | 84.5 | 70.7 |

## A.2 Theoretical Error Analysis

Consider the source and target distribution $S$ and $T$ over a family of classifiers $F$ in hypothesis space $\mathcal{H}_\mathcal{H}$. Denote $\epsilon_S(F) = \mathbb{E}_{(x,y)\sim S}|F(x) - y|$ as the risk of a hypothesis $F \in \mathcal{H}_\mathcal{H}$ in distribution $S$. For two hypotheses $F_1, F_2 \in \mathcal{H}_\mathcal{H}$ in distribution $S$, the disagreement could be calculated as follows:

$$\epsilon_S(F_1, F_2) = \mathbb{E}_{(x,y)\sim S}|F_1(x) - F_2(x)|. \tag{1}$$

The notions in distribution $T$ are similar. We denote the ideal hypothesis as $F^* = \arg\min_F \epsilon_S(F) + \epsilon_T(F)$. The probabilistic bound of the target risk $\epsilon_T(F)$ of hypothesis $F$ is given by the source risk, ideal joint hypothesis risk and domain discrepancy risk, as follows:

$$\epsilon_T(F) \leqslant \epsilon_S(F) + [\epsilon_S(F^*) + \epsilon_T(F^*)] + |\epsilon_S(F, F^*) - \epsilon_T(F, F^*)|. \tag{2}$$

For heuristic search, we have $F = G + H$, where $F$, $G$ and $H$ belong to the same hypothesis space $\mathcal{H}_\mathcal{H}$. Also the goal of $F$ and $G$ is the same, thus the ideal hypothesis $F^* = G^*$. Meanwhile, since both $G$ and $F$ could classify source samples correctly, the differences on source domain are little, which means $\epsilon_S(G) = \epsilon_S(F)$. Besides, $G$, $G^*$, $F$ and $F^*$ could achieve well-performed results on source domain. Then the disagreement on source domain could be expressed as $\epsilon_S(G, G^*) = \epsilon_S(F, F^*) = 0$.

On the way transferring to the ideal hypothesis, the heuristic function $H$ models domain-specific parts on both S and T. The domain-specific parts could also be approximately calculated by the parts in $G$ but not exists in $G^*$. Then the two parts could contain the overlap obtain domain-specific parts, resulting positive cosine similarity, as follows:

$$(G(x) - G^*(x)) \cdot H(x) = \|G(x) - G^*(x)\| \|H(x)\| cos(\alpha) \geqslant 0, \tag{3}$$

where $cos(\alpha) \in (0, 1)$ and $\alpha$ denotes the angle between $G(x)$ and $H(x)$. Then we could calculate the risk on target domain as follows:

$$
\begin{aligned}
\epsilon_T(F, F^*) &= \mathbb{E}_{(x,y)\sim T}|F(x) - F^*(x)| \\
&= \mathbb{E}_{(x,y)\sim T}|F(x) - G^*(x)| \\
&= \mathbb{E}_{(x,y)\sim T}|G(x) - G^*(x) + H(x)| \\
&= \mathbb{E}_{(x,y)\sim T}\sqrt{(G(x) - G^*(x) + H(x))^2} \\
&= \mathbb{E}_{(x,y)\sim T}\sqrt{(G(x) - G^*(x))^2 + (H(x))^2 + 2*(G(x) - G^*(x))H(x)} \quad (4) \\
&\geqslant \mathbb{E}_{(x,y)\sim T}\sqrt{(G(x) - G^*(x))^2 + (H(x))^2} \\
&\geqslant \mathbb{E}_{(x,y)\sim T}\sqrt{(G(x) - G^*(x))^2} \\
&= \mathbb{E}_{(x,y)\sim T}|G(x) - G^*(x)| \\
&= \epsilon_T(G, G^*).
\end{aligned}
$$

Similar to Eqn. 2, the target risk with the generator $G$, *i.e.*, $\epsilon_T(G)$ could also be calculated as follows:

$$
\begin{aligned}
\epsilon_T(G) &\leqslant \epsilon_S(G) + [\epsilon_S(G^*) + \epsilon_T(G^*)] + |\epsilon_S(G, G^*) - \epsilon_T(G, G^*)| \\
&= \epsilon_S(F) + [\epsilon_S(F^*) + \epsilon_T(F^*)] + |\epsilon_T(G, G^*)| \\
&\leqslant \epsilon_S(F) + [\epsilon_S(F^*) + \epsilon_T(F^*)] + |\epsilon_T(F, F^*)| \\
&= \epsilon_S(F) + [\epsilon_S(F^*) + \epsilon_T(F^*)] + |\epsilon_S(F, F^*) - \epsilon_T(F, F^*)|
\end{aligned}
\tag{5}
$$

Thus $G$ could restrain the error upper-bound to a lower value, which shows the superiority of heuristic domain adaptation.

(a) Classification accuracy     (b) Domain separation accuracy

Figure 3: Classification and domain separation accuracy on C→R.

Table 3: Parameter $M$ on Domainnet for MSDA.

| value of $M$ | $M=1$ | $M=2$ | $M=3$ | $M=4$ | $M=5$ | $M=6$ | $M=7$ |
|---|---|---|---|---|---|---|---|
| Clipart | 63.0 | 63.7 | 63.9 | 64.0 | 64.0 | 64.1 | 64.1 |
| Infograph | 25.5 | 25.7 | 25.9 | 26.0 | 26.3 | 26.3 | 26.2 |

## A.3 Experiment Analysis

We also calculate the cosine similarity in the training process as shown in Figure 1. The results show that the similarity could be reduced in the training process, but it is hard to result in zero. This shows that the domain-specific and domain-invariant representations are hard to be orthogonal. Besides, the results without heuristic initialization mainly different from HDA at the early stages in the training process. While the final values of cosine similarity with and without initialization are similar.

The results of heuristic loss on different heuristic subnetwork numbers could be found in Figure 2. Results show that multi-heuristic subnetworks could result in large heuristic loss in initialization. While the final heuristic losses are similar, on behalf of the destination.

The results with independence constraint are shown in Table 2. The results show that adding nongaussianity constraint on the framework of HDA brings negative effects. Since directly restricting the nongaussianity discrepancy could also affect the nongaussianity of each value of $F(x)$ and $G(x)$, which brings difficulties to the changes of nongaussianity in $F(x)$ and $G(x)$.

We also analyze the Number of $M$ in MSDA, as shown in Table 3. The results reach the highest when $M = 6$, and it seems the performance does not degrade with a larger $M$. So task such as MSDA contains more domain properties, and tends to need a larger $M$. But $M = 3$ could still guarantee well-performed results on arbitrary number of domains.

In experiment of Figure 2 in the paper, we choose the UDA settings with task of Clipart $\rightarrow$ Real as the task. The classification accuracy and domain separation results are shown in Figure 3. The task is chosen since clear validation parts are formulated in Domainnet. We build the classifier the same network structure with the discriminator, but the input features do not include the GRL layer. The final classification accuracy is 0.811 for source domain of Clipart, while 0.574 for target domain of Real. More importantly, the higher average accuracy of separating $H(x^s)$ and $H(x^t)$ means $H(x)$ could be more easily separated compared with $G(x)$. This validates that more domain properties in$H(x)$.