[Reviews · NeurIPS 2020]

Review 1

Summary and Contributions: The paper presents a method inspired by heuristic search algorithms (such as A*) for the problem of Domain Adaptation (DA). The authors claim that to achieve domain invariant representations, one must explicitly model domain-specific characteristics. The ideal representation is considered as the goal and the intermediate domain-specific representations are regarded as the distance from the current representation to the ideal one. When the heuristic representations are near zero the terminal state is reached. To achieve this goal the authors propose a domain adaptation network made of a fundament network F and a heuristic network H (that is possibly divided into several subnetworks). Transferable representations are the result of F(x) - H(x) = G(x). To construct and reduce the domain-specific characteristics, the authors propose three constraints on the representations: (1) Similarity - limit the initial value of the heuristic representations based on the cosine similarity between G(x) and H(x); (2) Independence - make G(x) and H(x) independent components in the representation; (3) Termination - the range of heuristic representations H(x) should be near zero. Item (1) is achieved by initializing the fundament network parameters close to zero and items (2) and (3) are achieved by applying an L1-norm on H(x) termed heuristic loss. The overall loss term includes heuristic loss, classification loss, and transfer loss. An upper bound for the error on the target set was presented in the paper and the method was demonstrated on the challenging DomainNet and Office-home datasets. The main contributions of the paper are (1) model domain invariant and domain-specific information inspired by heuristic search processes, (2) propose HADA a framework for domain adaptation, and (3) state-of-the-art performance on three domain adaptation tasks: Unsupervised DA, multi-source DA, and semi-supervised DA.

Strengths: The paper has several strengths: 1. The method presented in the paper tries to model domain-specific characteristics as well as domain-invariant representation. Although not novel, this is an interesting approach and is less common in the current field. 2. The proposed approach is general and can be applied to many DA setups as was shown in the paper. Also, it seems rather simple to implement or add to current methods. 3. The method was demonstrated on the challenging datasets Office-home and DomainNet and achieved good results on both compared to baseline methods. 4. The theoretical insight appears sound and corresponds with similar derivations seen in this field.

Weaknesses: I would like to get clarifications about the following: 1. Based on the intuition from heuristic search the authors chose to subtract the representation of H(x) from F(x). This is one approach. Have you tried other approaches as well, for example, using another NN over concatenated representations with the gradient reversal layer applied only on the fundament network? 2. Line 118 states that each sub-network tends to model local domain-specific property. Why? Can it be shown (even empirically)? Do they model different properties? And if that is indeed correct why a summation over the sub-networks' representations is the right thing to do? 3. The authors published their code and deserve compliments on that. However, the paper lacks in presenting several implementation details. Some I was able to find in the code. For example, what networks were used for the fundament network and heuristic network? How the hyper-parameters were chosen? Was early stopping applied? If so, on which part of the dataset (train, val, test) and on which domain? I would like to get answers to these questions and in general more details. From the code, it seems that early stopping was done based on the target-test set. If that is indeed the case, in my opinion, it is very problematic in an unsupervised setup.

Correctness: The method appears to generate good results on challenging datasets, however, as stated, some design choices that are not clear to me. Also, I would like to understand better the experimental setup before I can pass it.

Clarity: The paper is well written and easy to follow. I found a few minor typos: - line 110 the word "the" appears twice. - Table 4, R->S column, 3-shot is not consistent with the rest of the columns. - In table 1 in the appendix, line 5 should be g(n) instead of f(n). - line 24 in the appendix should be h(m).

Relation to Prior Work: The authors addressed prior works and nicely described how their approach is different.

Reproducibility: No

Additional Feedback: The authors motivate their method by heuristic search algorithms. However, to me, it seems a bit artificial. I wasn't convinced how strong this relationship truly is (even after reading the supplementary material). I think that the method is legitimate without this motivation. --------------------------------------------------------------------------------------------------------- The authors addressed my concerns adequately and it appears that they addressed other reviewers' comments as well. In my opinion, this is a decent paper with a rather unique solution that should be accepted therefore, I decided to raise my score from 5 to 6.


Review 2

Summary and Contributions: The authors use a heuristic search approach to look for domain invariant and domain specific representation to address a variety of domain adaptation settings. Taking cues from A* search, the authors design a heuristic network as an ensemble of multiple subnetworks each capturing domain-specific properties to learn domain-specific representation, while a domain invariant network learns invariant representation. Using constraints redoing non-gaussianity using Kurtosis measure and termination heuristic, the search for an optimal transfer function is architected.

Strengths: + The paper theoretically shows an upper bound on the error over the generative function to indicate the search for an optimal function is possible with low error. + Experimental results demonstrate the ability of the proposed approach over semi-supervised and unsupervised DA, and also indicates how the number of subnetworks chosen affects model performance.

Weaknesses: There are, however, a few unanswered questions. 1. Are the comparative results statistically significant? 2. In the ablation study, it is indicated that larger number of sub-networks (M) would be difficult to optimize. Is this true in the multi-source DA setting as well? If so, how does the choice of M affect the performance when there are different number of domain-specific properties within each domain? 3. What are the limitations of the proposed solution?

Correctness: Yes

Clarity: Yes

Relation to Prior Work: Yes

Reproducibility: Yes

Additional Feedback: Authors have adequately addressed my concerns in their rebuttal.


Review 3

Summary and Contributions: This paper explores to explicitly separate domain-specific characteristics from existing representations by a unified heuristic function. The heuristic function is achieved/enhanced by multiple subnetworks to ensure the accurate construction of domain-specific properties. This heuristic function is further integrated into adversarial domain adaptation, which formulate the framework of heuristic adversarial domain adaptation (HADA). Experimental results show that HADA could achieve state-of-the-art results on unsupervised domain adaptation, multi-source domain adaptation and semi-supervised domain adaptation.

Strengths: 1. It solves domain adaptation problem from a heuristic search perspective and provides a unified manner/framework to achieve domain-invariant feature learning. 2. Clear motivation and problem definition. 3. Satisfying performance on three different domain adaptation tasks.

Weaknesses: 1. Authors highlight “HADA could achieve lower error bound” several times in both the main paper and supplementary, but I am very confused by this statement. What is the intuitive meaning of that? In addition, how is this property related the superiority of HADA, any comparison? 2. Experiment wise, there are not enough ablation studies to support the superiority of using heuristic search theory. Besides, this paper has used multiple loss items. It would be helpful to know which one contributes more. 3. Typo. n-th at Line-104.

Correctness: Yes.

Clarity: Yes.

Relation to Prior Work: Yes.

Reproducibility: Yes

Additional Feedback: I'm curious what is the performance If we discard the heuristic search and incorporate some commonly used disentangle techniques to separate different properties. -------------------------------------------------------------------------------------- Authors addressed my concerns during rebuttal period.

[Author Response · NeurIPS 2020]

Table 1: Comparison of GRL applied to fundament network (FADA) and HADA

| UDA | A→C | A→P | A→R | Avg | MSDA | Clipart | Infograph | Painting | Avg | SSDA($1_{shot}$) | R→C | R→P | P→C | Avg |
|---|---|---|---|---|---|---|---|---|---|---|---|---|---|---|
| FADA | 55.3 | 71.3 | 77.5 | 68.5 | FADA | 61.5 | 23.5 | 53.9 | 45.7 | FADA | 69.5 | 66.7 | 70.0 | 67.6 |
| HADA | **56.8** | **75.2** | **79.8** | **70.9** | HADA | **63.9** | **25.9** | **56.1** | **47.9** | HADA | **71.7** | **67.1** | **72.8** | **69.5** |

Figure 1: $|H^k(x)|$

Figure 2: $G(x)$ and $H^k(x)$

Figure 3: $H^k(x)$ and $H^{k'}(x)$

Table 2: $M$ in MSDA

| HADA | Clipart | Infograph |
|---|---|---|
| $M=1$ | 63.0 | 25.5 |
| $M=2$ | 63.7 | 25.7 |
| $M=3$ | 63.9 | 25.9 |
| $M=4$ | 64.0 | 26.0 |
| $M=5$ | 64.0 | 26.3 |
| $M=6$ | 64.1 | 26.3 |
| $M=7$ | 64.1 | 26.2 |

**Reviewer1**. 1. Network structure. We apply gradient reversal layer (GRL) to the single fundament network (FADA).
Some results of FADA could be found in Table 1, which performs worse than HADA by a large margin. Besides, we
also try to concatenate $F(x)$ and $H(x)$ as the input of a one-layer neural network, and applying GRL to $F(x)$, but the
results are poor. For example, on A→C on Office-Home, its accuracy is 30.1 compared to 56.8 by HADA.

2. Sub-network. Each $H^k$ is initialized with random weights generated from Gaussian with different variance, resulting
in different output ranges in the training process, as shown in Figure 1. It could also ensure distinguished local
domain-specific properties captured by different $H^k$, as shown in Figure 2 and 3 . $H^k(x)$ can effectively capture rich
domain-specific properties, evidenced by the low similarity between $H^k(x)$ and $G(x)$, *i.e.*, $\cos(G(x), H^k(x)) \approx 0$, as
shown in Figure 2 . Moreover, $H^k(x)$ contains different local property, since $H^k(x)$ and $H^{k'}(x)$ ($k \neq k'$) is orthogonal
or negatively correlated, *i.e.*, $\cos(H^k(x), H^{k'}(x)) \leq 0$, as shown in Figure 3. Therefore, the aggregation of $H^k(x)$
could express more diversified domain specific properties encoded by each sub-network.

3. Experiments. The experiment settings and hyper-parameters follow [30][33]. Dataset separation on Office-Home
follows [33], and the separation on Domainnet follows [30]. In tasks of UDA, SSDA and MSDA, the total number of
iteration is fixed. Though the model might reach higher results during training, we still record the final performance
when reaching the highest number of iteration for fair comparison. So we emphasize that early stopping is not used.

4. Motivation. The tasks of heuristic search and HADA are quite different, but heuristic search inspires us to devise
both the heuristic network and termination constraint. To clarify the relationship, we will elaborate the connection
between the two in the revised paper and supplementary.

**Reviewer3**. 1. Statistical significance. We report each result as the average accuracy on three trials. By calculating the
std, we find that the performance gain by HADA is statistically significant. In Table 3, HADA achieves 47.9±0.40,
which outperforms comparisons by large margins.

2. Number of $M$. Results of different $M$ for MSDA are shown in Table 2. The results reach highest when $M=6$, and
it seems the performance does not degrade with a larger $M$. So task such as MSDA contains more domain properties,
and tends to need a larger $M$. But $M=3$ could still guarantee well-performed results on arbitrary number of domains.

3. Limitation. Separating domain-specific and domain-invariant parts without extra information could hardly achieve
perfect separation in real situations. It is possible to introduce some extra domain-specific knowledge to enhance the
separation. Also the structure of heuristic network could be enhanced by Neural Architecture Search (NAS).

**Reviewer4**. 1. Bound. The heuristic function could estimate and explicitly reduce the remaining domain-specific parts.
The explicit reduction could result in more effective reduction of domain properties, with lower error bound. The bound
is analyzed in many recent DA methods such as [33][56]. Lower error bound provides better theoretical guarantee for
better model, as validated by experiments.

2. Ablation study. Our main contributions are heuristic sub-networks, similarity initialization, and termination loss. We
tried HADA without each function in Table 2 in the paper. The performance without termination is poor, showing that
the termination loss is more important compared with similar initialization and multiple sub-networks. The classification
and adversarial loss constitute the basic framework of domain adaptation, which should not be removed.

3. Separation. Most existing separation methods are designed by statistic analysis, which seems to be less effective in
real situations. DSN [6] separates the representations by input image reconstruction, but the improvement over DANN
[17] is little (from 90.3 to 91.2 on Synth Digits to SVHN). We achieve significant improvement over DANN (from 57.6
to 70.9 on Office-Home).

[Meta-Review · NeurIPS 2020]

The paper addresses domain adaptation from the perspective of heuristic search, modeling explicitly the domain-specific characteristics. Transferrable representations were extracted from a fundamental network and a heuristic network, with the latter decomposed into several subnetworks. Three constraints were proposed on the transferrable representations: similarity, independence, and termination. The method is shown effective on challenging datasets such as office-home, with superior performance over several state of the art. Overall, this paper provides an interesting idea using heuristic search. The reviewers raised several concerns on experiments including ablation study. They have been addressed well by the rebuttal, and can be incorporated to the final version of the paper. I think this novel approach deserves publication at NeurIPS.